# Optimal CNN–Hopfield Network for Pattern Recognition Based on a Genetic Algorithm †

**Fekhr Eddine Keddous** and **Amir Nakib** *

Laboratoire LISSI, University Paris Est Creteil, 94000 Creteil, France; fekhr-eddine.keddous@univ-paris-est.fr
* Correspondence: nakib@u-pec.fr
† This paper is an extended version of our paper published in the proceedings of the 2021 IEEE International Parallel and Distributed Processing Symposium Workshops (IPDPSW), Portland, OR, USA, 17–21 June 2021.

**Abstract:** Convolutional neural networks (CNNs) have powerful representation learning capabilities by automatically learning and extracting features directly from inputs. In classification applications, CNN models are typically composed of: convolutional layers, pooling layers, and fully connected (FC) layer(s). In a chain-based deep neural network, the FC layers contain most of the parameters of the network, which affects memory occupancy and computational complexity. For many real-world problems, speeding up inference time is an important matter because of the hardware design implications. To deal with this problem, we propose the replacement of the FC layers with a Hopfield neural network (HNN). The proposed architecture combines both a CNN and an HNN: A pretrained CNN model is used for feature extraction, followed by an HNN, which is considered as an associative memory that saves all features created by the CNN. Then, to deal with the limitation of the storage capacity of the HNN, the proposed work uses multiple HNNs. To optimize this step, the knapsack problem formulation is proposed, and a genetic algorithm (GA) is used solve it. According to the results obtained on the Noisy MNIST Dataset, our work outperformed the state-of-the-art algorithms.

**Keywords:** convolutional neural networks (CNN); recurrent neural network (RNN); Hopfield neural network; genetic algorithm; knapsack problem; classification





## 1. Introduction

In the last decade, convolutional neural networks (CNNs) have become the standard methods for pattern recognition and image analysis. Indeed, they have provided highly successful advances in image classification problems [1–3]. CNNs are auto-encoders, which means that they encode or extract features automatically with any specific fitting [4,5]; this characteristic has increased their utilization in computer vision tasks, such as semantic segmentation tasks [6–9], image retrieval [10–12], and object detection [13–15]. However, one of main problems regarding their use in industrial systems comes from the computation time and the use of memory resources. This is because classic CNN-based architectures have at least one fully connected layer (FC) depending on the architecture's depth [16,17]. Figure 1 illustrates some popular examples of CNN models, where it can be clearly seen that the FC layer(s) contains most of the parameters of the network [1]; for instance, AlexNet [18] has about 64 million learnable parameters, and 91.56% of these parameters belong to the last three FC layers of the model; the same observation applies to the Lenet5 [19], VGG16 [1], and VGG19 [1] models. The majority of learnable parameters are in their FC layers, with 89.18%, 89.58%, and 86.11%, respectively, out of the total number of parameters that can be learned.

In the literature, some authors, such as those of [20], proposed the replacement of the FC layer with an associative memory bank. Indeed, this use of memory overcame the problems related to backpropagation; the features extracted from the CNN layers were

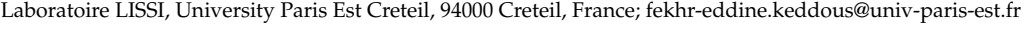

stored in the memory and then recovered at the inference phase. However, the storage capacity of the memory scheme remains a serious problem to be solved.

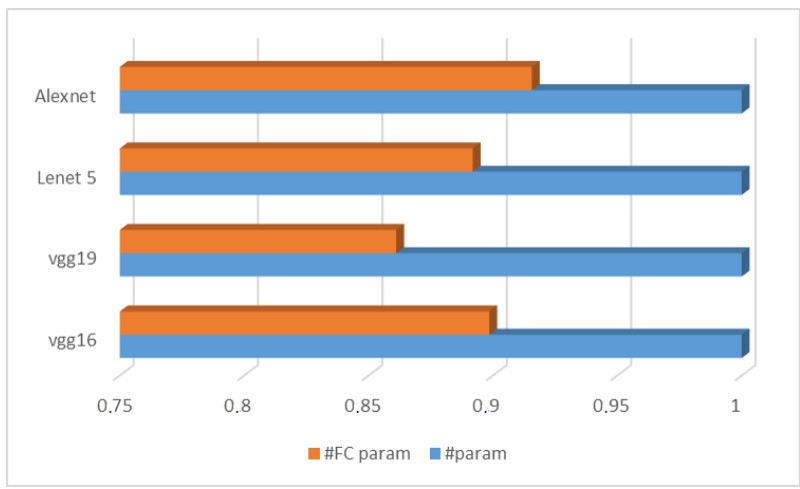

**Figure 1.** Ratio of the number of FC parameters to the total number of parameters in some well-known CNN models.

To deal with this problem, many authors have proposed alternatives to the classical neural network learning rules, such as in [21–23], and they have provided valuable information on the properties of attention heads in transformer architectures [24]. The contributions of this paper can be summarized in the following:

- New hybrid CNN–RNN (recurrent neural network) architecture and approach and its optimization.
- Inference acceleration of a CNN-based architecture.
- Formulation of the optimization of the architecture as a knapsack problem.

The architecture uses a pretrained CNN classification model to extract feature maps from the input images. The use of an associative memory bank (the Hopfield neural network) allows the replacement of the fully connected layer and, therefore, the large number of trainable weights (parameters) while preserving the performance. In order to increase the storage capacity and minimize the effects caused by the spurious states, several Hopfield networks are used in parallel.

In the following, in Section 2, we present an overview of the framework, while in Sections 3 and 4, the classical Hopfield model and a detailed description of the new formulation of the problem are presented. In Sections 5 and 6, the dataset used for the experimentation and the discussion of the obtained results are presented. Finally, the paper is concluded in Section 7.

## 2. General Description of the Method

In this section, an overview of the architecture of the proposed method in its two phases—the training phase and the inference phase—is presented. Then, an architecture based on multiple parallel HNNs, which replaces the fully connected layer of the CNN, is discussed, and the strategy for distributing the patterns among a set of HNNs in the parallel architecture is analyzed.

*Feature Extraction*

As shown in Figure 2, a pretrained CNN model is used for feature extraction. These features come from the last layer before the dense (fully connected) layers.

During the training phase, the architecture uses a pretrained CNN classification model to extract feature maps from an input image. The process consists of:

1. Extraction of the class-specific feature set: a set of all features of an image.

2. Averaging of the pixels' gray level from the class-specific feature set.
3. Conversion of the training patterns into binary patterns and distribution over *K* parallel Hopfield networks.

In the inference phase, the same pretrained CNN is used to extract the feature maps from the input image. Then, the binary pattern of the input image is sent to all HNNs. The correct network among all of the networks (which converges to a state that is very close to the input state) is selected based on the comparison of the states of the networks with the input pattern and the number of changes that each one has produced. Then, the network that has produced the fewest changes in the input pattern is selected.

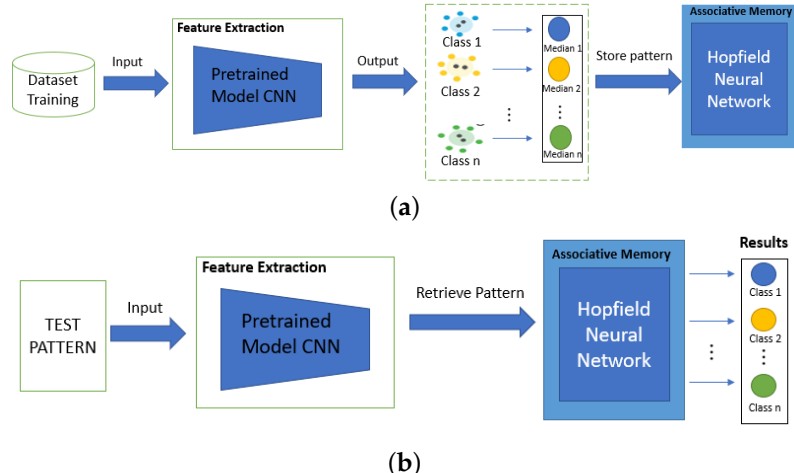

(**a**)

(**b**)

**Figure 2.** Overview of our hybrid CNN–RNN architecture. (**a**) training process; (**b**) inference process.

## 3. Recall of the Hopfield Neural Network

One of the well-known recurrent neural networks is the Hopfield network (HNN) [25–29]. This NN can be seen as an associative memory and has been applied in a large number of applications, such as classification [23,30], optimization [31–33], image processing [34], control [35], and solving blind detection problems [36,37].

A Hopfield network consists of a set of interconnected neurons *N* that update their activation values asynchronously and independently of other neurons. A neuron *i* is characterized by its state $S_i = \pm 1$. The principle of HNNs is to save binary patterns of the form $\{+1, -1\}^N$, and then to use a rule, called Hebb's rule, to learn them. In the inference phase, these patterns are then predicted via a noisy input vector. Its robustness to noise is very interesting in several kinds of applications. The "energy" of the Hopfield networks is defined by:

$$E = -\frac{1}{2} \sum_{i,j}^{N} S_i S_j w_{ij} \tag{1}$$

where $w$ is the weight associated with neurons *i* and *j*. $S_i$ is the state of neuron *i*. This quantity is considered as a Lyapunov function; it remains stable or decreases when the network states are updated. The HNN will converge to a local minimum in the energy function under repeated updating. So, the optimal values of the weights are those that minimize the energy function. Furthermore, the theoretical storage capacity of the HNN, under the assumption of the stability of all the patterns, is defined by [25]:

$$P_{max} = \frac{N}{4 \ln N} \tag{2}$$

where $P_{max}$ denotes the maximum number of uncorrelated patterns that can be stored in the *N*-neuron recurrent network [38]. Each pattern stored corresponds to a local minimum of the energy defined in (1). Recently, many works proposed new learning rules and new energy functions that improved Hopfield networks' properties. The storage capacity was

about $0.138 \times N$ [32,33,39]. In [40], $N$ patterns could be stored when the learning rate was not related to Hebb's rule. The use of new energy functions, such as interaction functions of the form of $F(x) = x^n$, as in [21,41], provided a storage capacity proportional to $N^{n-1}$. A modern energy function based on interaction functions of the form of $F(x) = exp(x)$ yielded a storage capacity of $2^{N/2}$ in [22]. In [24], the authors proposed a new update rule that ensures rapid global convergence, and they generalized the energy function of Demircigil [22] to continuous patterns; thus, the HNN became differentiable and could be integrated into deep learning architectures. In this case, the storage capacity of this model became proportional to $c^{\frac{N-1}{4}}$ (for c = 1.37 and c = 3.15).

## 4. Knapsack Model for Pattern Recognition

The key task of our approach is to optimize the storage of the different patterns. To deal with this problem, it is formulated as an assignment optimization problem or a knapsack problem [42]. This optimization problem is a well-known combinatorial optimization problem in which the objective is to find the optimal object from a set of items. The traditional 0–1 knapsack problem is defined by $N$ items, where $N = \{1, 2, 3, ..., n\}$ is the set of items. Each item has a weight, $w_n$, and a value, $p_n$. Since one knapsack can store only a maximum capacity or weight of $W$, the goal is to find the items that can be packed into the knapsack to maximize the packing. The formulation for this traditional 0–1 knapsack is given in (3). The binary decision variable, $x_n$, takes the value of one if the item is packed into the knapsack and zero otherwise. Therefore, the problem can be formulated as follows:

$$\begin{cases} \text{Maximize} \sum_{i=1}^{n} x_i p_i \\ \text{subject to,} \sum_{i=1}^{n} x_i w_i \leq W \\ \quad x_i \in \{0, 1\} \end{cases} \tag{3}$$

Then, the Hopfield network can be seen as a knapsack that can hold a set of items—in our case, patterns to be stored. Each pattern has a weight $w_i$ corresponding to the absolute energy of the pattern $x^\mu$ computed by (1) and a value $p_i$ that can be defined by measuring the similarity between the pattern and the other patterns. The capacity (or total weight) of the knapsack $W$ is defined by:

$$W = max\{|E_1|, |E_2|, ..., |E_n|\} \times P_{max} \tag{4}$$

The estimation of $P_{max}$ can be made from the upper bound of the theoretical capacity of the HNN computed by (2). In order to define the values of $n$ distinct items (patterns), one of the common similarity measures can be used. To select the most dissimilar (orthogonal) patterns among the patterns in the dataset, principal component analysis (PCA) or a similar method can be applied [43,44].

### 4.1. Similarity Measures

To compute the $n$ values of the knapsack, a similarity/orthogonality between our training patterns is measured. The similarity measure [45] is a distance that represents the features of the patterns. If the distance is small, two patterns are very similar, while a large distance means that they are more orthogonal. In this paper, the *cosine* similarity is considered. It is used to calculate the similarity between vectors:

$$\cos(\theta) = \frac{x^\mu \cdot x^\pi}{\|x^\mu\| \cdot \|x^\pi\|} \tag{5}$$

where $x^\mu$ and $x^\pi$ are two pattern vectors. The value of $\cos(\theta)$ lies in the interval $[-1, 1]$, where the value $-1$ means that the two vectors are exactly opposite; the value of 1 means that they are exactly equal, and the value of 0 means that they are orthogonal, while the values within the range $[0, 1]$ indicate similarity or dissimilarity. Furthermore, if elements of the vectors are binary and not bipolar ($-1$ and 1), the value of $\cos\theta$ is in the range $[0, 1]$,

so the value 0 means that the two vectors are orthogonal, while the value 1 indicates that they are exactly the same.

### 4.2. Setting of the Genetic Algorithm

In this paper, we consider a genetic algorithm [46,47] to solve this problem. This choice is motivated by the no-free-lunch [48] theorem and the success of genetic algorithms in solving knapsack problems [49–51]. However, a comparison of different metaheuristics for solving this problem is still in progress. The flowchart of the proposed genetic-algorithm-based approach is shown in Figure 3.

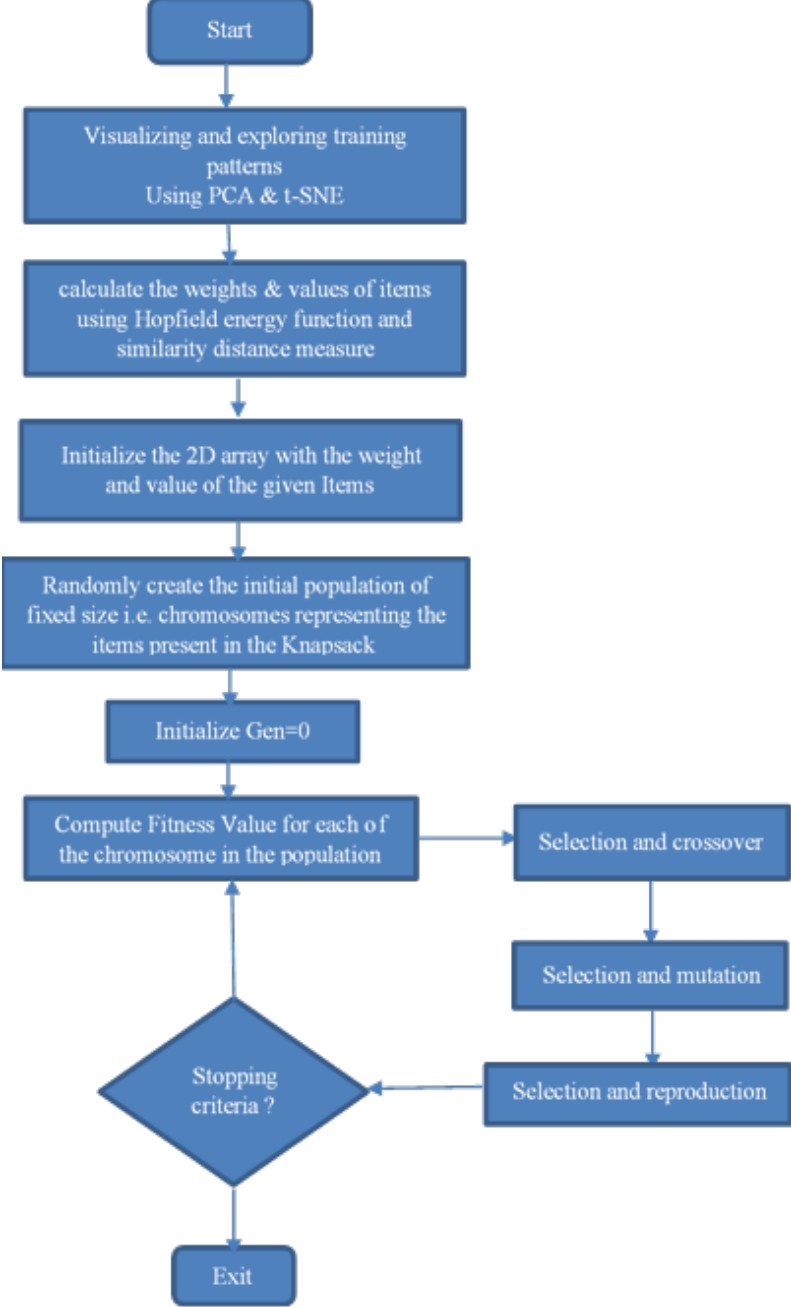

**Figure 3.** Flowchart of the proposed approach [52].

The input parameters are the dimension of the problem $N$, the population size $POP$, the maximum number of generations $ITER$, the crossover rate $CR$, and the mutation rate $MUT$. The algorithm starts with a random population. Individuals are binary vectors

evaluated by a fitness function defined in (6). The stopping criterion is defined by the maximum number of generations. The best solution $\vec{x}^*$ is the best individual of the last population. To solve the formulated problem, binary coding is used, where each chromosome is a sequence of bits: 0 or 1. The chromosome can be represented in an array whose size is equal to the number of the items. Each gene indicates whether an item is included in the knapsack ('1') or not ('0'). Regarding genetic operators, the roulette-wheel selection, the half-uniform crossover, and the classical bitflip mutation operator were used.

The fitness value of each chromosome is defined by the total values (profits) of the items included in the knapsack, being careful not to exceed the capacity of the knapsack. It can be noticed that a knapsack represents an HNN and the items represent the patterns stored in it.

When the total volume reached by a chromosome is greater than the capacity of the knapsack, a repair operation is performed. This involves reversing one of the (randomly selected) genes and rechecking the capacity. This repair operation is only performed once. If the individual cannot be repaired, it is eliminated. The proposed fitness function is defined by:

$$fitness(\vec{x}) = \begin{cases} \sum_{i=1}^n x_i p_i & if \sum_{i=1}^n x_i w_i \leq W \\ 0 & Otherwise \end{cases} \tag{6}$$

Since the number of chromosomes in each generation and the number of generations are fixed, the complexity of the program only depends on the number of items that can potentially be placed in the knapsack.

Figures 4 and 5 illustrate the relationship between the genetic algorithm and the CNN at the training phase and inference phase, respectively.

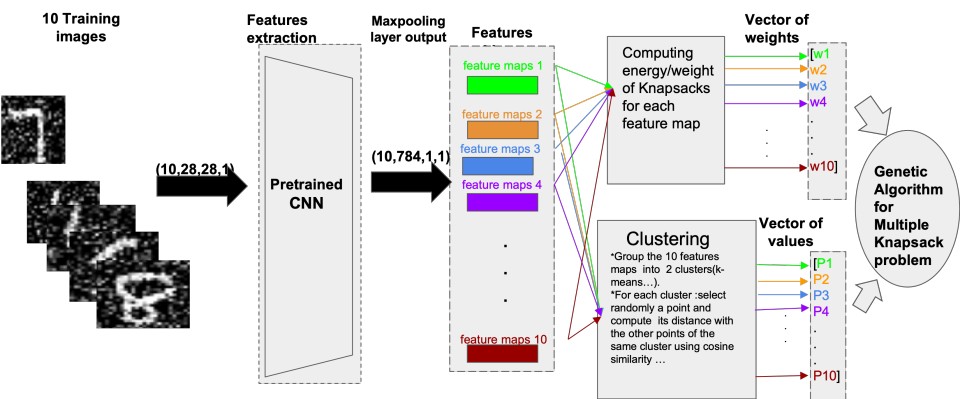

**Figure 4.** Illustration of the relationship between the genetic algorithm and the CNN at the training phase.

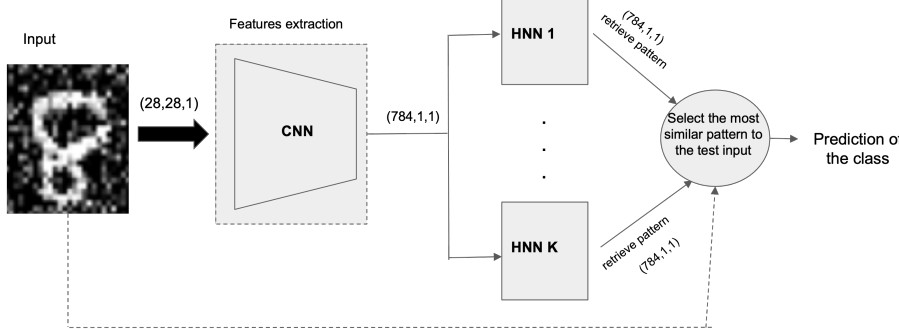

**Figure 5.** Illustration of the relationship between the genetic algorithm and the CNN at the inference phase.

### 4.3. The Optimal Number of Knapsacks (K)

To increase the storage capacity of the HNN, a modular network architecture of knapsacks (Hopfield networks operating in parallel) was developed. The idea is to split the training dataset into subsets and distribute them among the different networks. Then, a selection procedure is applied to choose the best network to store the pattern with the requirement of retrieving it with high accuracy. Of course, it is necessary to point out a some problems related to the proposed architecture.

Figures 6 and 7 show the proposed neural architecture in the training phase and the recovery phase, respectively. In order to optimize the number of associative memories that can be used to efficiently store the patterns, first, a naive approach is used, which can be the basic framework or the upper bound of the final solution. Then, the proposed approach based on a genetic algorithm is presented.

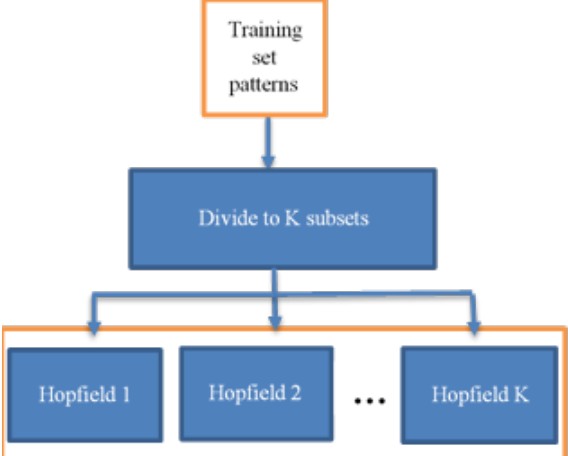

**Figure 6.** Block diagram of the multi-RNN architecture proposed in the training phase [52].

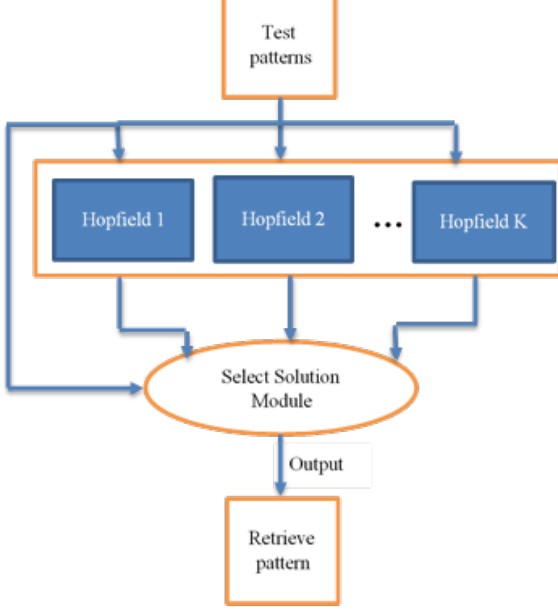

**Figure 7.** Block diagram of the multi-RNN architecture proposed in the inference phase [52].

### 4.3.1. Naive Approach

In this case, the accuracy of the classification is used to measure the performance of the model. Indeed, it is the main criterion for defining the number *K* of networks that we will consider. To this end, we first store all of the training patterns in a knapsack (Hopfield network). Depending on the accuracy, the number of knapsacks is increased by 1.

Moreover, the training patterns are randomly divided into $K$ subsets, and each of these subsets is then trained on these networks. In the inference phase, the selection system compares the output of these networks to the test vector until a candidate network converges. If these networks fail to properly recover all patterns, we increase $K$ and (randomly) redistribute the patterns. This procedure is repeated until the desired prediction accuracy is achieved.

### 4.3.2. Heuristic Approach

The other main reason for the low memory capacity of the classical Hopfield network is the linear combination of the patterns [53]. In this paper, a heuristic method based on a genetic algorithm is proposed for assignment of patterns to multiple parallel HNNs.

### 4.4. Pattern Distribution

To distribute the training patterns over the $K$ parallel Hopfield networks, we use the multiple knapsack problem, which is a variant of the knapsack problem. We consider a set of patterns $N = \{1, 2, 3, ..., n\}$, which we load into $K$ multiple knapsacks, each with a capacity of $W$. Each item (pattern) $j \in N$ is characterized by its weight $w_j$, its value $P_j$, and its decision variable $x_{ij}$, which is equal to 1 if pattern $j$ is loaded into knapsack $i$ and is equal to 0 otherwise. Then, the problem is to find $K$ disjoint subsets of $N$ that maximize the total value of the selected patterns. The problem can be formulated as follows:

$$
\begin{aligned}
&\text{Max} \sum_{i=1}^{K} \sum_{j=1}^{n} P_j x_{ij} \\
&\text{subject to} \sum_{j=1}^{n} w_j x_{ij} \leq W, \\
&\text{with} x_{ij} \in \{0,1\}, j \in \{1, ..., n\}, i \in \{1, ..., K\}
\end{aligned}
\tag{7}
$$

### 4.5. Knapsack Selection

In the inference phase, the Hamming distance [54] is used to quantify the difference between the input test pattern and the output states of the networks at each update iteration. The network that has changed the least is selected as the winning network that contains the pattern we are looking for.

Consider $T$ and $S$ as two patterns; $T$ is the test pattern and $S$ is the stored average/median pattern. The Hamming distance between them is a function $DH[T, S]$, which is the number of bits that are different between $T$ and $S$.

At the end of the inference phase, we compute the associated class by matching the test pattern with the output pattern of the selected network by computing and comparing the similarities using distance metrics such as the cosine similarity, Euclidean distance, Jaccard index, etc. In this paper, the cosine similarity is considered as follows:

$$
CosineSimilarity(T, S) = \frac{T \cdot S}{\|T\| \|S\|}
\tag{8}
$$

Then, the stored core pattern can be retrieved based on the following equation:

$$
L = \text{Argmin} CosineSimilarity(T, a_k^i)
\tag{9}
$$

where $a_k^i$ is the $i^{th}$ stored binary pattern for the $k^{th}$ class. For each test pattern $T$ in a test image, the algorithm computes the difference between $T$ and each stored pattern $a_k^i$ ,$i \in \{1, ..., z\}$, $k \in \{1, ..., n\}$ in class $k$ using the formula in (8). Then, (9) is used to acquire the set of patterns $R$ that have the minimal distance from $T$. In Algorithm 1, the whole algorithm is presented.

---

**Algorithm 1** Classification Algorithm

---

1: **Input:** Test pattern t;
   Stored patterns $a_k^i, i \in \{1, \dots, z\}, k \in \{1, \dots, n\}$
2: **Output:** Class label $l$ of test pattern.
3: **Initialize:** $L \leftarrow \varphi$;
4: $L = \arg\min_{a_k^i | i \in 1 \dots z, k \in 1 \dots n} CosSimilarity(t, a_k^i)$;
5: **Return** $l = label(first(L))$;

---

## 5. Datasets and Benchmarks

The experiments mainly focused on a noisy version of a commonly used handwritten digit dataset, Noisy MNIST Dataset [55]. This is the same as the original MNIST dataset [19], except for the added noise. Each version contains 10 classes with a total of 70,000 gray-level images (60,000 training images and 10,000 test images) with an image size of $28 \times 28$.

We consider three different versions of each dataset: the first with added white Gaussian noise (AWGN) in Figure 8, the second with reduced contrast with white Gaussian noise (Contrast) in Figure 9, and the third with motion-blur noise (Motion) in Figure 10.

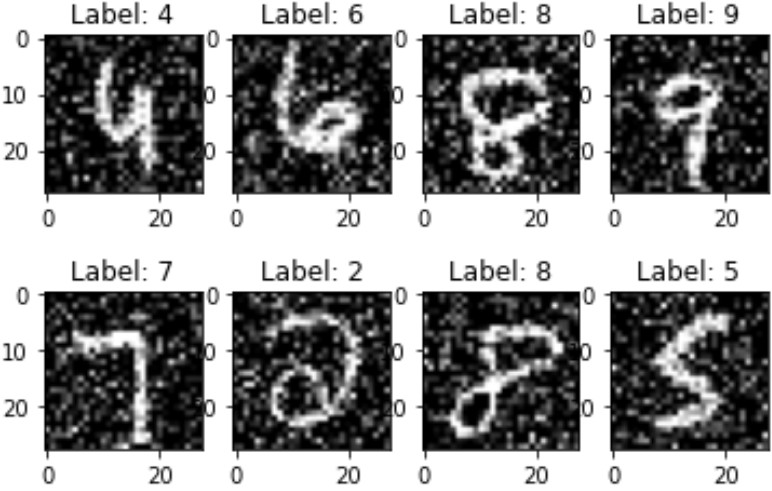

**Figure 8.** Image samples from the MNIST dataset with AWGN.

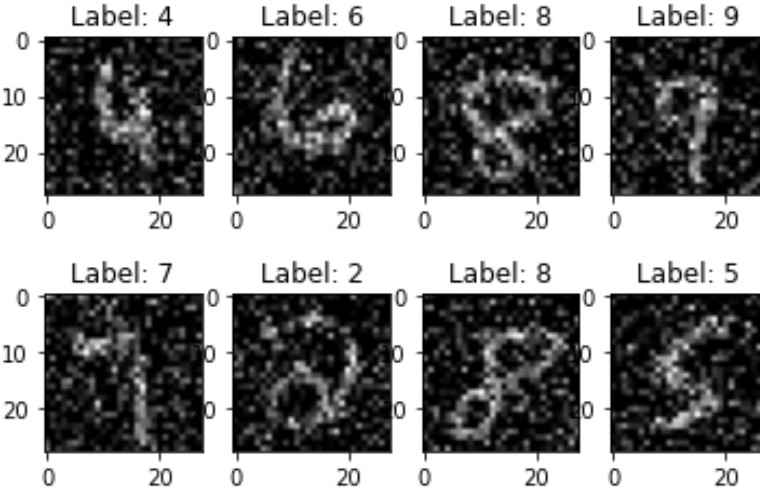

**Figure 9.** Image samples from the MNIST dataset with AWGN+Contrast.

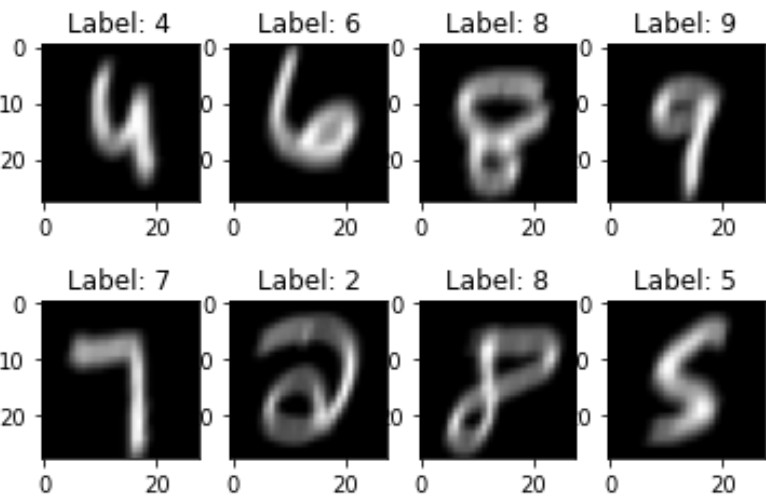

**Figure 10.** Image samples from MNIST dataset with motion-blur noise.

## 6. Results and Discussion

In this section, we present and illustrate the performance of the proposed method.

### 6.1. Implementation Details

#### 6.1.1. Pretrained CNN LeNet5-Like Model

We used a modified LeNet5 as a pretrained CNN model (trained on the Noisy MNIST Dataset) for feature extraction. LeNet5 [19] is a simple model composed of two convolutional layers, two average pooling layers, and three fully connected layers. The LeNet5 architecture is a popular network that is known to work well on digit classification tasks. This model gives a high accuracy on MNIST datasets; nevertheless, the model suffers from the problems of both high variance, which represents the overfitting, and high bias, which represents underfitting of the model. In order to minimize both of them and handle the noisy version of MNIST, some changes in the architecture need to be made:

- Maxpooling instead average pooling for reducing variance;
- Data augmentation to enhance the accuracy;
- A batch normalization layer after every set of layers (convolution + maxpooling and fully connected) to stabilize the network;
- Addition of dropout layers with a hyperparameter of 40% after the pooling layers;
- Addition of some connected layers;
- Addition of two more convolution layers with the same hyperparameters, and the number of filters in the convolutional layers was significantly increased from 6 to 32 in the first two layers and 16 to 100 in the next two layers to handle bias.

This new model was trained on 60,000 training images (split into a training and validation set) for the three versions of noisy MNIST using the stochastic gradient descent algorithm; the loss function used was the categorical cross-entropy. The total number of trainable parameters was 324,858, of which 200,514 parameters belonged to the last three FC layers (61.72% of the parameters were in the three FC layers). The activation function used was Relu (rectified linear units). In our CNN–RNN architecture and during the training phase, we chose a single representative pattern for each class. It was calculated by averaging the brightness of the pixels from the class-specific feature set; therefore, we only stored 10 patterns instead of 60,000 training images.

The features were pooled from the second maxpooling layer before the fully connected dense layers of the modified LeNet-5, which had $1 \times 1600$ dimensions. These 10 feature vectors were normalized and binarized with a threshold value of $p_{th} = 86$, then distributed among the networks.

6.1.2. Defining Weights and Values of the Knapsack

The size of each network is $N = 784$ neurons, and the energy of each pattern $w_n$ is calculated with (1). Therefore, we rely on McEliece's rule [38] to estimate the capacity of the Hopfield network and, therefore, the upper bound of the capacity of our knapsack (Hopfield network) by applying (2). Then, the bias of the training patterns is measured from the average of each class, and the correlation $c$ is measured using the average covariance between successive patterns. In order to define the vector of values $P$ of the knapsacks, it is necessary to identify the patterns that are correlated. To do this, one can calculate cosine similarity matrix and perform a principal component analysis (PCA). Based on the PCA's 2D projection of the three noisy MNIST datasets in Figures 11–13, respectively, one can define a pair of similar objects; for instance, if we take the reduced contrast and AWGN version (Figure 11), the pairs [item1,item2] , [item0,item6], [item4,item9], [item7,item8], and [item3,item5] could be chosen. The obtained weights and value for three noisy MNIST versions are presented in Tables 1 and 2.

**Table 1.** Weights for the three noisy MNIST versions.

|  | Weights |
|---|---|
| Contrast | 2764268, 3030404, 3083368, 3496748, 3519312, 3528504, 2773188, 3214824, 3508152, 4099424 |
| AWG | 3336168, 3425580, 3184744, 3424400, 3675664, 3653880, 3135436, 3355700, 3730228, 4031064 |
| MotionBlur | 3126420, 3498624, 3133432, 3450532, 3199344, 3243000, 3072960, 3299396, 3441172, 3760432 |

**Table 2.** Values for the three noisy MNIST versions.

|  | Values |
|---|---|
| Contrast | [1,0.403,1,0.605,1,1,0.491,1,0.449,0.722] |
| AWGN | [1,1,0.45,1,0.649,0.406,0.437,0.365,0.384,1] |
| MotionBlur | [1,1,0.38,1,0.442,0.588,0.52,1,0.552,0.578] |

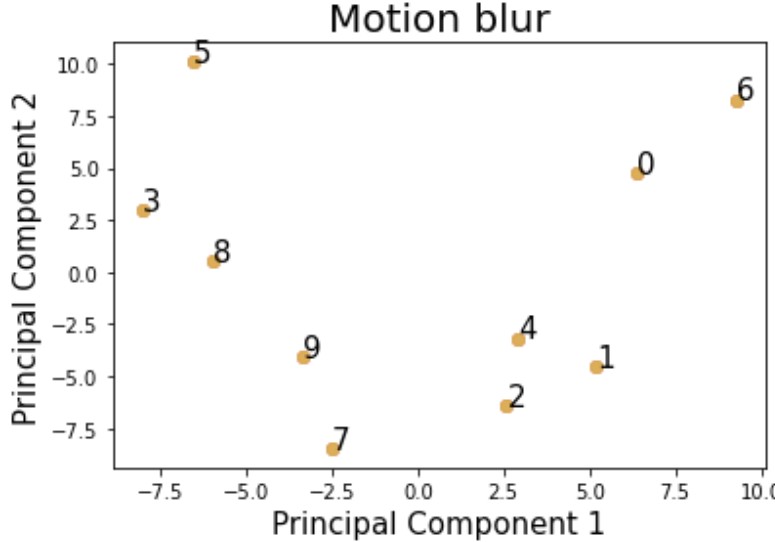

**Figure 11.** Two-dimensional principal component analysis projection of motion-blur noise.

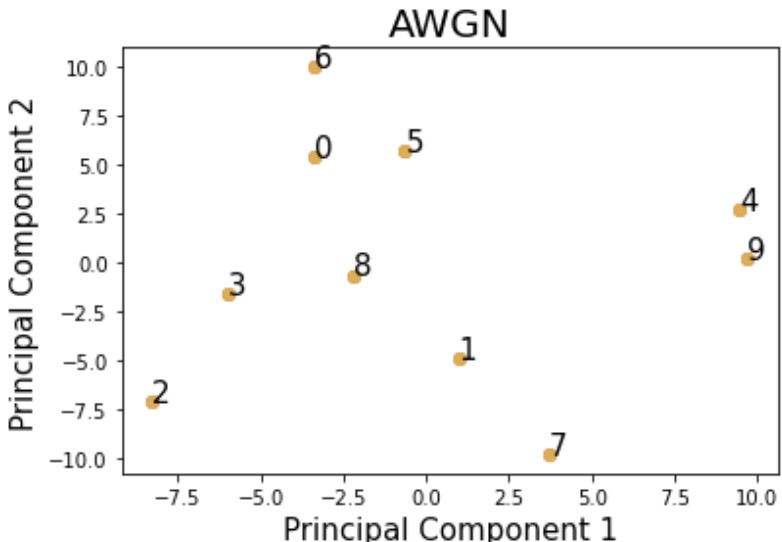

**Figure 12.** Two-dimensional principal component analysis projection of AWGN noise.

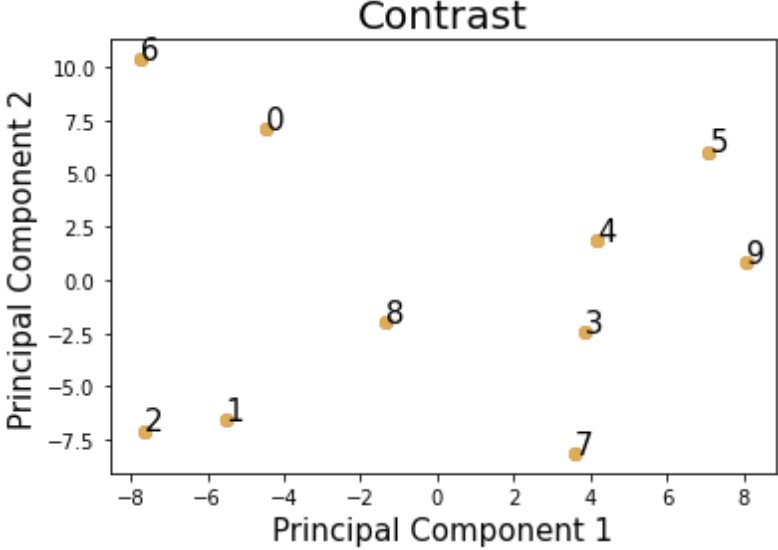

**Figure 13.** Two-dimensional principal component analysis projection of AWGN+Contrast noise.

6.1.3. Genetic Algorithm

The genetic algorithm used to solve the formulated problem and its implementation were performed via the Pymoo Framework [56]. The different settings of the parameters are presented in Table 3.

The quality of each chromosome was determined by the value of the fitness function. The fitness of each chromosome was defined by the sum of the benefits of the items included in the knapsack while making sure that the capacity of the knapsack was not exceeded. Given the stochastic nature of the GA, we performed 20 independent runs of the algorithm.

During the inference phase, our tests were carried out on 10,000 test images (patterns); we extracted their features and compared them to all of the outputs provided by the networks. Therefore, the network with the closest pattern iwass selected. This network would return the label when it reached a stable state. Figures 14–16 show the evolution of the fitness over the first 20 generations in the cases of one, two, or three knapsacks. The algorithm found the best solution after 20 generations. Items that were chosen for each knapsack with the max fitness found are reported in Table 4 for the three dataset versions.

**Table 3.** Parameters of the genetic algorithm.

| Parameters | Values |
|---|---|
| Dimension of the problem (N) | K |
| Number of runs | 20 |
| Number of generations | 20 |
| Population size (POP) | 100 |
| Max number of generations (ITER) | 100 |
| Mutation rate (MUT) | 0.1 |
| Crossover rate (CR) | 0.9 |

**Table 4.** The chosen items for different numbers of knapsacks with the max fitness found.

| | Contrast | | AWGN | | MOTION | |
|---|---|---|---|---|---|---|
| | Max. Fit. Found | Items Chosen | Max. Fit. Found | Items Chosen | Max. Fit. Found | Items Chosen |
| One knapsack | 5 | 1,3,5,6,8 | 4 | 1,2,4,10 | 3.588 | 1,4,6,8 |
| Two knapsacks | 7.267 | 1,5,9,10 3,4,6,7,8 | 5.942 | 1,2,3,4 5,6,7,10 | 6.680 | 5,6,7,8,10 1,2,4,9 |
| Three knapsacks | 7.669 | 1,2,3,4,5 6,7,8,9 10 | 6.691 | 1,2,3,4 5,6,7,8 9,10 | 7.06 | 1,2,3,4 5,6,7,8 9,10 |

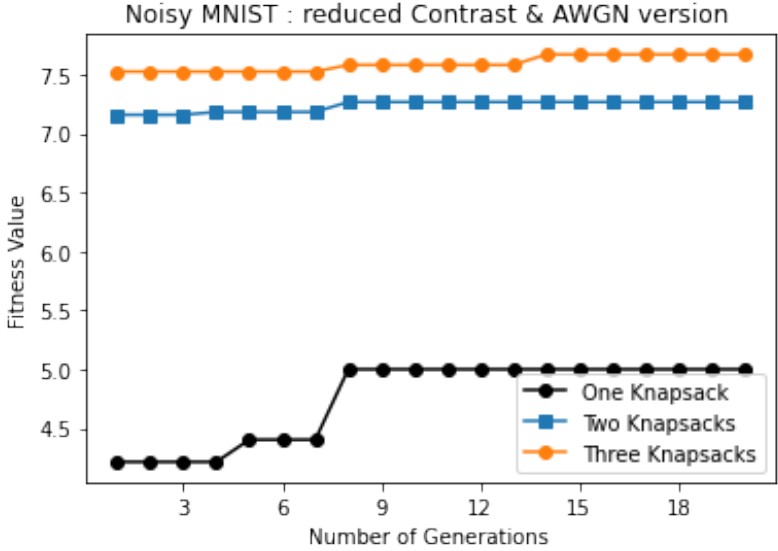

**Figure 14.** GA fitness vs. the number of generations for AWGN+Contrast noise.

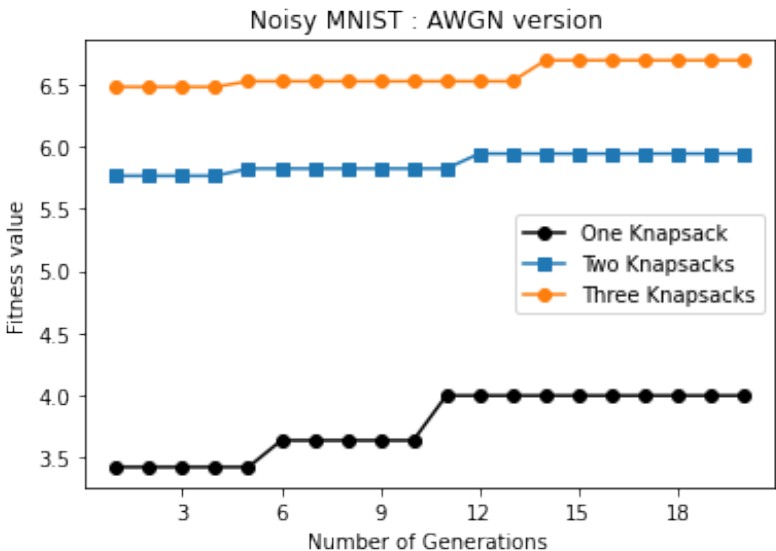

**Figure 15.** GA fitness vs. the number of generations for AWGN.

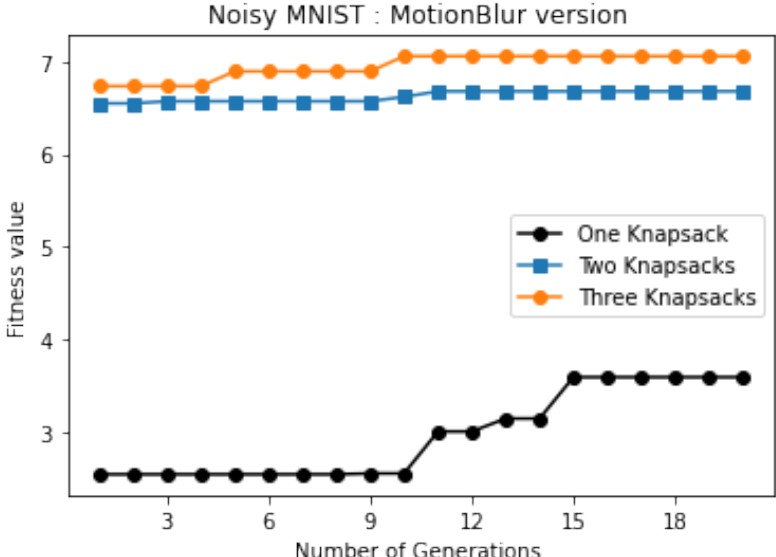

**Figure 16.** GA fitness vs. the number of generations for motion-blur noise.

### 6.2. Evaluation of the Performance

Herein, the performance of the proposed architecture for the classification task with respect to the number of sets of *K* Hopfield networks is presented. The results in Figures 17–19 are based on the three versions of the Noisy MNIST Dataset. The proposed architecture is compared with the naive approach.

The increased number of parallel networks used to store training patterns had a direct impact on the classification accuracy. In fact, in our approach, we needed about two to three networks to achieve the best performance on the three versions of the Noisy MNIST Dataset. The performance remained relatively stable even after using more than three networks.

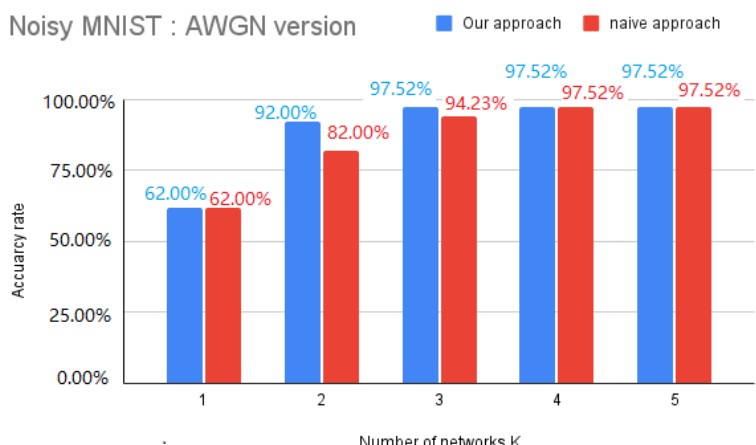

**Figure 17.** Classification's accuracy on the MNIST dataset with added white Gaussian noise.

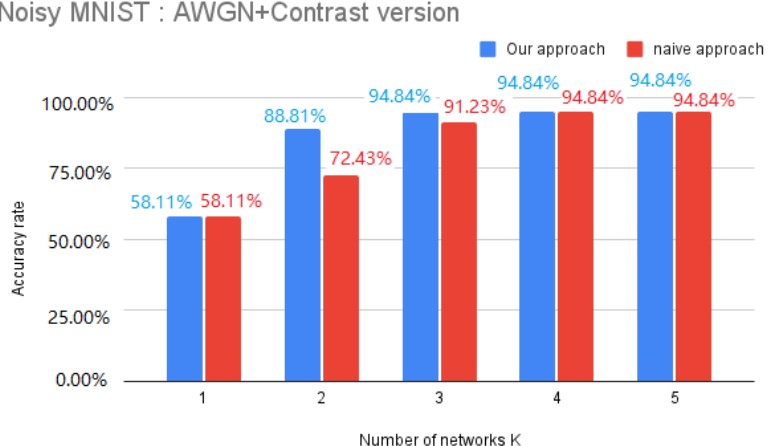

**Figure 18.** Classification's accuracy on the MNIST dataset with reduced contrast + AWGN.

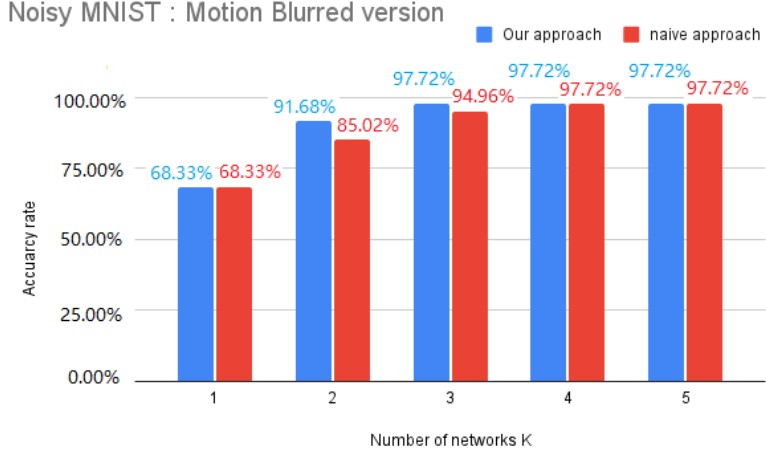

**Figure 19.** Classification's accuracy on the MNIST dataset with motion-blurred noise.

The total number of trainable parameters was 324,858, of which 200,514 parameters belonged to the last three FC layers, which means that 61.72% of the parameters were in the last three FC layers. Our architecture achieved remarkable results in terms of accuracy and memory requirements, as shown in Table 5. The inference time performance (CPU Intel Core i3) was also kept and improved in our hybrid architecture. In order to classify a single

test AWGN image, Lenet took 0.8368 ms, while our hybrid version only took 0.7954 ms; the same observations were observed on the other noisy versions of the datasets (see Table 6).

**Table 5.** Performance (accuracy) comparison between the Lenet-like architecture (with FC layers) and our hybrid Lenet–Hopfield architecture.

| Model | # of Parameters | Accuracy AWGN | Accuracy Motion | Accuracy Contrast |
|---|---|---|---|---|
| Lenet5-Like (with FC Layers) | 324,858 | 97.12% | 96.50% | 93.82 % |
| Our approach (parallel networks) | **124,344** | **97.52%** | **97.72%** | **94.84%** |

**Table 6.** Performance (inference time in ms) comparison between the Lenet architecture (with FC layers) and our hybrid Lenet–Hopfield architecture.

| Models | AWGN | Motion | Contrast |
|---|---|---|---|
| Lenet | 0.8368 | 0.7228 | 0.9427 |
| Our approach | **0.7954** | **0.7020** | **0.8632** |

Table 7 illustrates the accuracy obtained by the proposed hybrid architecture using the Hopfield network. One can see that the results obtained exceeded those of the state of the art by 0.75% in the case of added AWGN noise, by 0.54% in the case of added motion noise (motion blur), and 0.28% in the case of contrast+AWGN noise, and our architecture gave the best classification accuracies of 99.18%, 99.74%, and 97.53%, respectively.

**Table 7.** Comparison of the classification accuracy on the three versions of the Noisy MNIST Dataset.

| Models | AWGN | Motion | Contrast |
|---|---|---|---|
| Dropconnect [55] | 96.02% | 98.85% | 93.24% |
| Karki et al. [55] | 97.62% | 97.20% | 95.04% |
| PCGAN-CHAR [57] | 98.43% | 99.20% | 97.25% |
| Our approach (parallel networks) | **97.52%** | **97.72%** | **94.84%** |

## 7. Conclusions

In this paper, a CNN architecture was combined with an HNN for pattern recognition. The aim of this proposal was to reduce the number of parameters of the CNN (for a classification task) while increasing or at least keeping the same accuracy. It was pointed out that CNN models have a large number of weights concentrated at the fully connected layers, so we proposed the replacement of these layers while maintaining the performance.

Then, the main idea was to exchange those FC layers with associative memories (Hopfield neural networks): a new architecture composed of multiple networks in parallel, where the training patterns were split into subsets by applying a local search approach. Furthermore, the assignment problem was formulated as a knapsack problem and solved via a genetic algorithm. The efficiency of the proposed CNN–Hopfield architecture on the Noisy MNIST Dataset was demonstrated experimentally. In this work, it was shown that the increase in the storage capacity of the associative memory (Hopfield network) considerably improved the performance on the classification application.

The generalization of this approach to other applications and datasets is in progress.

**Author Contributions:** Data curation, F.E.K.; writing—original draft preparation, A.N. All authors have read and agreed to the published version of the manuscript.

**Funding:** This research received no external funding.

**Institutional Review Board Statement:** Not applicable.

**Informed Consent Statement:** Not applicable.

**Data Availability Statement:** Not applicable.

**Conflicts of Interest:** The authors declare no conflict of interest.

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
