# Peer review of "Optimal CNN–Hopfield Network for Pattern Recognition Based on a Genetic Algorithm†"

_algorithms, doi:10.3390/a15010011_

Round 1

Reviewer 1 Report

In the paper, the authors proposed to replace the FC layers of CNN with a Hopfield Neural Network. There are many points need to be improved.
1. Literature review should cover more relative studies such as some recent papers related to Hopfield network as follows.

     1.1 Shaowei Chen, Shujuan Yu, Zhimin Zhang and Yun Zhang, “A Novel Blind Detection Algorithm Based on Adjustable Parameters Activation Function Hopfield Neural Network”, Journal of Information Hiding and Multimedia Signal Processing, Vol. 8, No. 3, pp. 670-675, May 2017

    1.2 Yun Zhang, Zhimin Zhang, Ying Liang, Shujuan Yu, Li Wang and Yan Zhang, “Blind Signal Detection Using Complex Transiently Chaotic Hopfield Neural Network”, Journal of Information Hiding and Multimedia Signal Processing, Vol. 9, No. 3, pp. 523-530, May 2018

2. More advanced and state-of-the-art pre-trained models should be included.
3. More datasets should be added into experiments.
4. The experimental analysis of memory consumption and computational complexity is shallow. More analysis should be given.
5. The curves in Table 15 to Table 17 look the same. Are there any differences? 
6. The writing should be substantially improved.

Author Response

In the paper, the authors proposed to replace the FC layers of CNN with a Hopfield Neural Network. There are many points need to be improved.

1. Literature review should cover more relative studies such as some recent papers related to Hopfield network as follows.

Shaowei Chen, Shujuan Yu, Zhimin Zhang and Yun Zhang, “A Novel Blind Detection Algorithm Based on Adjustable Parameters Activation Function Hopfield Neural Network”, Journal of Information Hiding and Multimedia Signal Processing, Vol. 8, No. 3, pp. 670-675, May 2017

Yun Zhang, Zhimin Zhang, Ying Liang, Shujuan Yu, Li Wang and Yan Zhang, “Blind Signal Detection Using Complex Transiently Chaotic Hopfield Neural Network”, Journal of Information Hiding and Multimedia Signal Processing, Vol. 9, No. 3, pp. 523-530, May 2018

We considered the remark and added more the two references suggested please reference section in the revised version of the paper and 20 other references.

2
. More advanced and state-of-the-art pre-trained models should be included. We included the state-of-the-art models on the noisy MNIST, please see table 6.

3. More datasets should be added into experiments.
This task in work under progress.

4
. The experimental analysis of memory consumption and computational complexity is shallow. More analysis should be given.

We considered this remark and added more analysis.

5. The curves in Table 15 to Table 17 look the same. Are there any differences? 

In terms of values, they are not the same. One can see that the behavior of the architectures is the same in both cases.

6.
The writing should be substantially improved.

We considered this remark and corrected the writing.

Reviewer 2 Report

This work presents a method to decrease the parameters related to CNN levels by replacing them with another type of layer that exploits a memorization of the patterns without backpropagation issues.

The work is interesting and well written, however there are some aspects to be clarified. One of the major limitations of this work is that authors confronted with a dataset (MNIST) and a network model (LeNet) not currently in the state of the art which represent problems that are already known and solved.

Furthermore, the proposed approach seems to be linked to the comparison between all the elements of the dataset: it is not specified what happens, in terms of computability, when the number of images or classes to be recognized increases?

The conclusions must be extended: the authors should, for example, explain if and how the proposed method can be applied or extended to datasets of a different nature or to tasks other than the classification of the entire image, as done with MNIST.

Minor issues:

- Abstract: the first half is useless, it says things not related to the research performed in the paper

- Line 30: "FC layer (s) contains most of the parameters of the network" -> provide some examples, to differentiate normal parameters from the FC ones

- Line 35: "to be solved!" -> remove the exclamation

- the formula (1) should be explained better, even using a diagram or a real example, what does it practically measure?

- Fig.1: what is the unit of measurement of the X axis?

- Line 38: “we present ..” -> remove

- Line 51: sections 4 and 5 missing

- Line 57-61: the same concept is repeated 4 times

- Line 83: It is not clear from (3) what Pmax measures and how it can be practically calculated

- Line 150: another -> Another

- Par. 3.2: a diagram of a real example would help to understand the relationship between the genetic algorithm pattern (intended as a generated data structure) and the pattern of the training / test example that you want to 'simulate and include' in the knapsack

- Table 1 is illegible

- the reference section should be expanded

Author Response

This work presents a method to decrease the parameters related to CNN levels by replacing them with another type of layer that exploits a memorization of the patterns without backpropagation issues.

The work is interesting and well written, however there are some aspects to be clarified.

`One of the major limitations of this work is that authors confronted with a dataset (MNIST) and a network model (LeNet) not currently in the state of the art which represent problems that are already known and solved.

We agree that LeNet model is the state of the art for MNIST but for the noisy MNIST it is not the case. Please table 5 where we added the state-of-the-art models for this case.

Furthermore, the proposed approach seems to be linked to the comparison between all the elements of the dataset: it is not specified what happens, in terms of computability, when the number of images or classes to be recognized increases?

The complexity is linear function of the number classes (O(N) where N: is the number of classes) and the complexity is quadratic function of the size of the data set O(D^2/2) where D is the size of the data-set.

The conclusions must be extended: the authors should, for example, explain if and how the proposed method can be applied or extended to datasets of a different nature or to tasks other than the classification of the entire image, as done with MNIST.

We considered this remark and extended the conclusion

Minor issues:

- Abstract: the first half is useless, it says things not related to the research performed in the paper.

We considered this remark

- Line 30: "FC layer (s) contains most of the parameters of the network" -> provide some examples, to differentiate normal parameters from the FC ones

We considered this remark, please see the introduction

- Line 35: "to be solved!" -> remove the exclamation

We considered this remark

- the formula (1) should be explained better, even using a diagram or a real example, what does it practically measure?

We considered this remark and added explainations

- Fig.1: what is the unit of measurement of the X axis?

The ratio between the #parameters of the FC and the total #parameters.

- Line 38: “we present ..” -> remove We considered this remark

Round 2

Reviewer 2 Report

This new version of the paper certainly denotes improvements and the authors have generally answered the doubts raised by the reviewers, adding useful material to clarify many aspects. Future works should be clarified better, as well as the resolution of some figures.